# Characterization of Neurons Expressing the Novel Analgesic Drug Target Somatostatin Receptor 4 in Mouse and Human Brains

**DOI:** 10.3390/ijms21207788

**Published:** 2020-10-21

**Authors:** Angéla Kecskés, Krisztina Pohóczky, Miklós Kecskés, Zoltán V. Varga, Viktória Kormos, Éva Szőke, Nóra Henn-Mike, Máté Fehér, József Kun, Attila Gyenesei, Éva Renner, Miklós Palkovits, Péter Ferdinandy, István M. Ábrahám, Balázs Gaszner, Zsuzsanna Helyes

**Affiliations:** 1Department of Pharmacology and Pharmacotherapy, Medical School & Szentágothai Research Centre, Molecular Pharmacology Research Group, University of Pécs, H-7624 Pécs, Hungary; angela.kecskes@aok.pte.hu (A.K.); pohoczkykriszti@gmail.com (K.P.); viktoria.kormos@aok.pte.hu (V.K.); eva.szoke@aok.pte.hu (É.S.); kun.jozsef@pte.hu (J.K.); 2Centre for Neuroscience, University of Pécs, H-7624 Pécs, Hungary; kecskes.miklos@pte.hu (M.K.); nora.henn-mike@aok.pte.hu (N.H.-M.); istvan.abraham@aok.pte.hu (I.M.Á.); 3Department of Pharmacology, Faculty of Pharmacy, University of Pécs, H-7624 Pécs, Hungary; 4Institute of Physiology, Medical School & Szentágothai Research Centre, University of Pécs, H-7624 Pécs, Hungary; 5Department of Pharmacology and Pharmacotherapy, Semmelweis University, H-1089 Budapest, Hungary; varga.zoltan@med.semmelweis-univ.hu (Z.V.V.); peter.ferdinandy@pharmahungary.com (P.F.); 6HCEMM-SU Cardiometabolic Immunology Research Group, Department of Pharmacology and Pharmacotherapy, Semmelweis University, H-1089 Budapest, Hungary; 7ALGONIST Biotechnologies GmbH, A-1030 Wien, Austria; 8Institute of Physiology, Medical School & Szentágothai Research Centre, PTE-NAP Molecular Neuroendocrinology Research Group, University of Pécs, H-7624 Pécs, Hungary; 9Department of Neurosurgery, Kaposi Mór Teaching Hospital, H-7400 Kaposvár, Hungary; fehmate@freemail.hu; 10Bioinformatics Research Group, Genomics and Bioinformatics Core Facility, Szentágothai Research Centre University of Pécs, H-7624 Pécs, Hungary; gyenesei.attila@pte.hu; 11Human Brain Tissue Bank, Semmelweis University, H-1089 Budapest, Hungary; renner.eva@med.semmelweis-univ.hu (É.R.); palkovits.miklos@med.semmelweis-univ.hu (M.P.); 12Pharmahungary Group, H-6720 Szeged, Hungary; 13Department of Anatomy, Medical School, Research Group for Mood Disorders, University of Pécs, H-7624 Pécs, Hungary; 14PharmInVivo Ltd., H-7629 Pécs, Hungary

**Keywords:** SST_4_ receptor, RNA scope in situ hybridization, β-galactosidase immunohistochemistry, pain, mood regulation, somatosensory cortex, M-current

## Abstract

Somatostatin is an important mood and pain-regulating neuropeptide, which exerts analgesic, anti-inflammatory, and antidepressant effects via its Gi protein-coupled receptor subtype 4 (SST_4_) without endocrine actions. SST_4_ is suggested to be a unique novel drug target for chronic neuropathic pain, and depression, as a common comorbidity. However, its neuronal expression and cellular mechanism are poorly understood. Therefore, our goals were (i) to elucidate the expression pattern of *Sstr4*/*SSTR4* mRNA, (ii) to characterize neurochemically, and (iii) electrophysiologically the *Sstr4*/*SSTR4*-expressing neuronal populations in the mouse and human brains. Here, we describe SST_4_ expression pattern in the nuclei of the mouse nociceptive and anti-nociceptive pathways as well as in human brain regions, and provide neurochemical and electrophysiological characterization of the SST_4_-expressing neurons. Intense or moderate SST_4_ expression was demonstrated predominantly in glutamatergic neurons in the major components of the pain matrix mostly also involved in mood regulation. The SST_4_ agonist J-2156 significantly decreased the firing rate of layer V pyramidal neurons by augmenting the depolarization-activated, non-inactivating K^+^ current (M-current) leading to remarkable inhibition. These are the first translational results explaining the mechanisms of action of SST_4_ agonists as novel analgesic and antidepressant candidates.

## 1. Introduction

Somatostatin (SS) was originally isolated as growth hormone-inhibiting factor from sheep hypothalamus [1]. It is an inhibitory neuromodulator synthesized by interneurons [2], somatotropic cells of the pituitary, neuroendocrine cells of the gastrointestinal tract, and inflammatory cells [3]. SS inhibits several physiological processes, including endocrine, exocrine, vascular, and neuronal functions, but also pathophysiological mechanisms, such as pain and inflammation [4,5].

All of these effects are mediated through 5 G_i_ protein-coupled receptors (SST_1-5_, encoded by the *Sstr1-5* in mouse and *SSTR1-5* genes in human). Based on their structural similarities and synthetic agonist binding abilities, SST receptors are categorized into two classes, SRIF1 (SST_1_, SST_4_) and SRIF2 (SST_2_, SST_3_ and SST_5_) [6]. We previously demonstrated that SST_4_ activation mediates a broad range of anticonvulsive, anti-inflammatory, and analgesic actions without influencing hormone secretion [7,8,9,10]. Although anti-nociceptive, antidepressant and anxiolytic functions of SS were initially linked to SST_2_ and SST_3_ [11], we provided evidence for the importance of SST_4_ in these mechanisms [12,13]. 

SS decreases excitability and neurotransmitter release of cortical and hippocampal neurons by modulating several calcium and potassium channels [14]. It inhibits N-type calcium channels of granule cells in the dentate gyrus, modulates synaptic plasticity and suppresses epileptogenesis [15]. Besides, SS enhances potassium conductance, including G protein-coupled inward-rectifying potassium channels (GIRK) and voltage sensitive K_v_7 channels (M-current) leading to hyperpolarization and decreased excitability [16,17]. Although neuronal inhibition of SS via these ion channels is well-established, the receptorial mechanisms and signaling pathways necessary for drug developmental efforts are poorly investigated. Activation of SST_4_ in the central nervous system leads to M-current augmentation and epileptiform activity inhibition in hippocampal CA1 pyramidal cells [10]. In addition, SST_4_ activation increases potassium conductance via GIRK channels leading to consequent suppression of voltage-sensitive calcium channels in primary sensory neurons of the dorsal root ganglia (DRG) [18].

*Sstr4* expression in the rodent central nervous system has been poorly investigated [12,19,20]. In the rat, SST_4_ immunoreactivity was shown in olfactory bulb, cerebral cortex, hippocampus, and amygdala by a custom-made antibody [19]. However, specific antibodies for reliable determination of the SST_4_ protein are currently not available. Immunohistochemistry results were partially supported by radiolabeled in situ hybridization (ISH) and receptor autoradiography in the mouse brain [20]. These methods have several limitations, such as low sensitivity and lack of co-localization possibilities with neuronal markers to reveal cell-type specific expression patterns. We previously showed strong *Sstr4*-specific *lacZ* signal in the stress-related central and basolateral amygdala of *Sstr4*-deficient, *lacZ* knock-in (*Sstr4^lacZ/lacZ^*) reporter mice by β-galactosidase immunohistochemistry (β-Gal IHC) [12]. However, this platform is not appropriate to determine expression changes under pathophysiological conditions, investigate functions and co-localizations for cell-type specific characterization. The novel, ultrasensitive RNAscope ISH technology provides a great opportunity to overcome these problems [21].

There is a strong proof-of-concept that SST_4_ is a unique novel drug target for neuropathic pain, which could be also valuable to treat depression as a common comorbidity [8,9,22,23,24]. Non-peptide SST_4_ agonists have recently come in the focus of interest of pharma industry [25]. However, the expression profile of *Sstr4* and its function including signaling mechanisms are barely explored. Therefore, our aims were to (i) elucidate the expression pattern of *Sstr4*/*SSTR4* mRNA, and (ii) characterize neurochemically and (iii) electrophysiologically the *Sstr4*/*SSTR4^+^* neuronal populations in the mouse and human brains. 

## 2. Results

### 2.1. Sstr4 Expression in Several Pain- and Mood Regulation-Related Brain Regions

Strong *Sstr4* mRNA expression was detected by RT-qPCR in the prelimbic cortex, hippocampus and habenula, while moderate, but still remarkable levels were detected in the amygdala and primary somatosensory cortex of mouse brain micropunches (*N* = 5, Appendix A). In human post-mortem samples (*N* = 4–8) without major brain pathologies (medical history of patients are listed in Appendix A), stable *SSTR4* expression was found in several brain areas related to nociceptive processing and depression, with the highest mRNA levels were found in the amygdala (Appendix A). All primers used for RT-qPCR are listed in Appendix A.

In order to investigate the expression pattern of *Sstr4* at the cellular level, we used the *lacZ* knockout/knock-in reporter mice (*N* = 3), *Sstr4^lacZ/lacZ^* in which the full length *Sstr4* coding sequence was replaced by the bacterial *lacZ* gene. In accordance with the RT-qPCR results, intense *Sstr4*-specific *lacZ* signal was depicted by β-Gal IHC in the prelimbic cortex (PrL, Figure 1A, left panel), in the layer V of primary somatosensory cortex (S1, Figure 1E, left panel), in the CA1 of hippocampus (CA1, Figure 1F, left panel) and in the medial nucleus of habenula (MHb, Figure 1G, left panel). Supporting the β-Gal IHC, abundant endogenous *Sstr4* expression was found using RNAscope ISH (*N* = 4) in mouse PrL (Figure 1A, right panel), in the layer V pyramidal neurons of S1 (Figure 1E, right panel), in the CA1 (Figure 1F, right panel), and in the MHb (Figure 1G, right panel). Moderate *Sstr4* expression was revealed by both techniques in the basomedial amygdala (BMA, Figure 1B), in the core of central amygdala (CeC, Figure 1C), and in the basolateral amygdala (BLA, Figure 1D).

Negative controls for RNAscope were performed using a probes designed to bacterial *dapB* gene. They gave no detectable fluorescent signal on any channel (*N* = 2, Appendix A). RNAscope 3-plex mouse positive control probes were used to visualize the mouse housekeeping genes: RNA polymerase II subunit A (*Polr2a*), peptidyl-prolyl cis-trans isomerase B (*Ppib*) and polyubiquitin-C (*Ubc*) mRNA (*N* = 2; Appendix A). As negative tissue controls, there was also no detectable *Sstr4* signal in the CA1 and S1 derived from *Sstr4^lacZ/lacZ^* animals (*N* = 2, Appendix A, left panel). In parallel, no *lacZ* signal was depicted either in the CA1 or in the S1 (Appendix A, right panel) from C57BL6/J (WT) mice (*N* = 2) by β-Gal IHC providing evidence for the specificity of both methods.

Additionally, we tested two commercially available antibodies (PA3-208 and GTX70677) on WT and *Sstr4^lacZ/lacZ^* mice. We could not detect any specific SST_4_ immunoreactivity either by immunofluorescence or 3,3′-diaminobenzidine (DAB) staining (Appendix A).

### 2.2. Sstr4 Is Detectable Both in Nociceptive and Anti-Nociceptive Centers as Well as in Glutamatergic Excitatory Neurons of Brain Regions Involved in Mood Regulation

Next, we performed RNAscope ISH for *Sstr4* in the main centers of the primary nociceptive spinothalamic system in the mouse. *Sstr4* mRNA was detected in the *Rbfox3* (NeuN)-positive sensory neurons of the L5-DRG (*N* = 3) (Figure 2A), in the laminae V–VII of the spinal cord L5 (*N* = 3) (Figure 2B), *Rbfox3* (NeuN)-positive sensory neurons of trigeminal ganglia (TG, *N* = 3) (Figure 2C). In contrast, no remarkable *Sstr4* mRNA was found in the ventral posterolateral thalamic nucleus (VPL, *N* = 3) (Figure 2D). 

In the S1 cortex, *Sstr4* and *Vglut1* co-localization confirmed that *Sstr4* is strongly expressed in layer V pyramidal cells (Figure 3C), but low-copy *Sstr4* transcripts were also detectable throughout all other layers (Figure 3A,B). These findings suggest that the neuroanatomical basis for SST_4_-mediated influence on protopathic sensibility is given at all levels of the spinothalamic system except at the third-order neuron level in the VPL of thalamus.

Selected regions of the descending anti-nociceptive system were also stained for *Sstr4* mRNA in mice. The medial preoptic area of the hypothalamus (MPO, Figure 4A) as well as the in the ventrolateral periaqueductal gray matter (VLPAG, Figure 4B) express some *Sstr4* mRNA (Figure 4B arrow) not exclusively in *Gad1*^+^ neurons (Figure 4B, arrowhead). *Sstr4* transcripts were also detected in the noradrenergic (i.e., TH-immunoreactive cells) of the locus coeruleus (LC, Figure 4C). In the raphe magnus nucleus (RMg) few *Sstr4* mRNA transcripts were detectable, but did not co-localize with the serotoninergic cells (Figure 4D). The superficial laminae of the spinal dorsal horn contribute also to the descending pain control, and considerable *Sstr4* signal was detected here (Figure 4E). These neuroanatomical data support that the main centers of the descending pain control system may be influenced by SST_4_-related mechanisms.

Multiplex fluorescent RNAscope revealed cellular co-localization of *Sstr4* and *Vglut1* mRNA in several higher-order limbic areas implicated in multiple aspects of mood control, including the PrL (Figure 5A), BMA (Figure 5B), BLA (Figure 5D), CA1 (Figure 5E) and MHb (Figure 5F). Besides the abundant expression of *Sstr4* in *Vglut1^+^* excitatory neurons, few copies were also detectable in *Gad1^+^* inhibitory interneurons in all above mentioned regions. Additionally, the highest expression of *Sstr4* in *Gad1^+^* interneurons was found in the CeC (Figure 5C). Furthermore, *Chat^+^* cholinergic neurons in the MHb also showed *Sstr4* positivity (Figure 5G). 

### 2.3. Activation of the SST_4_ Receptor Decreases the Excitability of Layer V Pyramidal Neurons

Based on the expression data, we were interested whether the G_i_-coupled inhibitory SST_4_ receptors are functionally active. To investigate this in patch clamp experiments, we chose pyramidal cells of layer V of the S1, where *Sstr4* was abundantly expressed and tested whether the receptor activation by J-2156 alters cellular excitability. In agreement with the literature [26], J-2156 proved to be a potent and efficacious agonist in our SST_4_-linked G-protein activation assay (*n* = 3). It induced [^35^S]GTPγS binding in a concentration-dependent manner, its EC_50_ value was 92 nM (potency), the maximal activation was 267% (efficacy) (Appendix A). J-2156 significantly reduced firing frequency upon step current injection (250 pA) only in layer V pyramidal cells (*N* = 4, *n* = 10, *p* = 0.003; Figure 6E,F), but not in layer II–IV pyramidal neurons (*N* = 4, *n* = 12, *p* = 0.075; Figure 6C,D) or in interneurons (*N* = 3, *n* = 7, *p* = 0.79; Figure 6A,B). Since no specific SST_4_ antagonists are available, as a negative control we used *Sstr4^lacZ/lacZ^* mice, layer V pyramidal cells derived from these mice were not affected by J-2156 (*N* = 3, *n* = 12, *p* = 0.73; Figure 6G,H). The measured decrease in excitability of layer V pyramidal cells could be caused by a more hyperpolarized resting membrane potential or a decreased input resistance after J-2156 application. However, these parameters were unchanged in our experiments (resting membrane potential: ctrl: −70.4 ± 0.9 mV, J-2156-treated: −70.9 ± 0.95 mV; input resistance: ctrl: 69.3 ± 4.2 MΩ, J-2156-treated: 68.5 ± 4.5 MΩ). Individual cell morphology and soma location were verified by post hoc biocytin cell tracing (Appendix A). 

The mechanism attributed to the decreased firing frequency is depolarization-activated, non-inactivating potassium current (M-current). Accordingly, we examined M-current in S1 neurons following J-2156 administration. These experiments demonstrated that J-2156 significantly augmented M-current in layer V pyramidal neurons (*N* = 4, *n* = 11, *p* = 0.016; Figure 7C), but not in interneurons (*N* = 4, *n* = 6, *p* = 0.53; Figure 7A) or layer II–IV pyramidal neurons (*N* = 5, *n* = 11, *p* = 0.56; Figure 7B). 

### 2.4. SSTR4 mRNA Co-Localizes on VGLUT1-Expressing Pyramidal Neurons in Human Neurosurgical Cortical Tissue Samples

Similarly to *Sstr4* mRNA expression in the mouse somatosensory cortex, human *SSTR4* transcripts were shown in *VGLUT1*^+^ glutamatergic layer V pyramidal cells of the middle temporal gyrus (*N* = 2; Figure 8). Unfortunately, granular signal of lipofuscin accumulation resembles specific RNAscope signal, which we could not eliminate without quenching *SSTR4* or *VGLUT1* signals. Therefore, we showed both the lipofuscin signal (see arrows, Figure 8A,B) and specific *SSTR4* transcripts (see arrowheads, Figure 8A,B). RNAscope was validated by RNAscope 3-plex negative control probes designed to bacterial *dapB* gene giving no detectable fluorescent signal on any channel (*N* = 1, Appendix A). RNAscope 3-plex human positive control probes were used to visualize the human housekeeping genes: RNA polymerase II subunit A (*POLR2A*), peptidyl-prolyl cis-trans isomerase B (*PPIB*) and polyubiquitin-C (*UBC*) mRNA (*N* = 1; Appendix A). Lipofuscin signals are shown on all control images (see arrows, Appendix A).

## 3. Discussion

We provide here the first comprehensive evidence for (i) the expression pattern of SST_4_ in both the spinothalamic system and in the centers of descending anti-nociceptive pathway, (ii) the neurochemical characterization of the SST_4_-positive cells in several pain and mood-related mouse and human brain regions, and (iii) the functional relevance and molecular mechanism of action of SST_4_ activation. Using the ultrasensitive RNAscope technique, we showed that besides the major protopathic and anti-nociceptive centers, abundant expression mainly in glutamatergic excitatory neurons in the majority of the studied brain regions. However, *Sstr4* mRNA was also present in GABAergic inhibitory interneurons and cholinergic neurons in a few regions of the mouse brain. We show the first electrophysiological results for the inhibitory effect of SST_4_ activation via the augmentation of M-currents in layer V glutamatergic pyramidal cells of the mouse somatosensory cortex. These findings are essential to understand the role of SST_4_ in pain and mood disorders often occurring as comorbidities, and the mechanism of action of SST_4_ agonists, as novel analgesic and antidepressant drug candidates. These highlight the translational relevance of our neurochemical and electrophysiological findings in the mouse. For determining the expression pattern and presenting neurochemical characterization of *Sstr4*-expressing cells, we used the highly sensitive and reproducible fluorescent multiplex RNAscope technique [21]. This revealed that SST_4_ at the first (i.e., DRG and TG) and second (i.e., spinal dorsal horn) but not at the third-order neuron (i.e., VPL) level of the spinothalamic system may modulate the process of nociception. Anti-nociceptive centers, such as the MPO, VLPAG, and ventromedial medulla [27,28], as well as the LC [29], were found to contain *Sstr4*-expressing cells also. As in these centers, *Sstr4* was found, besides GABAergic cells, also in other neurons, including norepinephrinergic cells, a complex role of this receptor is predicted, which requires further, focused studies on anti-nociceptive centers. 

The *Sstr4* expression was found predominantly on *Vglut1*-positive glutamatergic excitatory neurons in the PrL, BMA, BLA, CA1, MHb, and in the layer V pyramidal neurons of S1. Meanwhile, we also identified *Sstr4*-expressing GABAergic interneurons in the CeC and cholinergic neurons in the MHb. 

We show that SST_4_ is present in numerous limbic brain areas that suggests its potential involvement in pain and mood control. The ratio of glutamatergic and GABAergic neuronal activity was shown to affect mood status both in rodent models and in human studies [30]. Our findings demonstrating that *Sstr4* is highly expressed in glutamatergic cells of the PrL suggest that its excitatory output is strongly inhibited by SS via SST_4_ activation. Meanwhile, based on the detected low copy *Sstr4* signal in GABAergic PrL neurons, we propose that SS, although to a smaller extent, may decrease the inhibitory influence of GABAergic interneurons on the pyramidal cells of the PrL. 

A similarly complex picture emerged in the amygdala. Although *Sstr4* transcripts were visualized principally in glutamatergic cells, some *Gad1*-expressing cells contained low number of *Sstr4* mRNA copies. The BLA harbors glutamatergic principal cells, which are controlled by a smaller population of GABAergic interneurons [31]. *Sstr4* was seen mainly in glutamatergic cells suggesting that principal cells receive an inhibitory modulatory input by SS that affects the control of the other nuclei of the amygdala via efferent BLA connections. This control, on the other hand, may be influenced by SS to some extent in an opposite way, since GABAergic neurons of the BLA were seen to express low copy *Sstr4* mRNA, too. The BMA was shown to contribute to the top-down control of anxiety states in rodents [32]. Here we found a strong *Sstr4* signal in BMA glutamatergic cells suggesting that SS potentially decreases the anxiolytic effect of BMA. The central nucleus is the main output of the amygdala that harbors GABAergic interneurons [33]. In our earlier work, we found that chronic stress in *Sstr4^lacZ/lacZ^* mice resulted in higher neuronal activity at the central amygdala, accompanied by an increased hypothalamus-pituitary-adrenal (HPA) axis response and depression-like phenotype in the tail suspension test [12]. The selective lesion of the central amygdala leads to decreased anxiety and depression level associated with blunted HPA axis response [34,35]. Based on these, we speculate that inhibition via SST_4_ on the GABAergic cells of the CeC may be required for the maintenance of normal HPA axis activity and mood level. However, further experiments are required to examine the neurochemical character of SST_4_-expressing interneuron subpopulations in the CeC.

Although habenular nuclei are overseen as mood control area, more recently the attention is turning to this neuroanatomical region [36,37]. Cholinergic neurons of the MHb also contain glutamate, that has been shown to contribute to the neurobiological background of nicotine and drug addiction and to mood control [38]. The selective inhibition of the MHb cholinergic system resulted in anhedonia-like behavior in the rat [39]. Here we show that these cholinergic neurons express *Sstr4* suggesting a potential role of SS in mood control via neuromodulation in MHb. 

A wide spectrum of limbic functions is attributed to the hippocampus. For instance, there is a well-known hippocampal inhibitory influence on the stress response of the HPA axis [40]. Interestingly, intra-hippocampal SST_4_ agonist treatment was shown to reduce the HPA axis activity and decrease depression-like behavior in mice [41]. Our current data suggest that this phenomenon may take place at least in part via SST_4_ in CA1 glutamatergic neurons.

Besides the allocortex, neocortical layer II and V S1 glutamatergic neurons were also found to contain *Sstr4* transcripts. All these transcriptional and functional data suggests that SS via SST_4_ modulates the somatosensory systems even at cortical level. Indeed, SS-containing interneurons were shown to innervate both layer II and V pyramidal cells in the cortex [42]. Low copy *Sstr4* mRNA signal was found in GABAergic interneurons of the S1, although this awaits further subgroup-specific neurochemical characterization.

In summary, our morphological data suggest that SS conveys a stronger inhibitory modulatory effect via SST_4_ on the glutamatergic and cholinergic neurotransmission in the limbic system and also in the S1. Meanwhile, SS acting via SST_4_ may inhibit GABAergic interneurons, as well result in reduced disinhibition. 

The present results show a good example for RNAscope being an excellent method to map expression profile of otherwise difficult-to-detect targets, such as SST_4_, similarly as previously published data [43]. Besides multiplexing necessary for co-localization studies, the other great advantage of this technique is that it provides an opportunity for exploring disease-related expression alterations, which is not possible with β-Gal IHC that only applicable in gene knockouts. Due to its unique Z probe pair design and signal amplification cascade strategy, RNAscope assures (i) detection at single mRNA molecule resolution in the full context of tissue architecture, (ii) successful hybridization of partially degraded RNA, (iii) high target specificity, (iv) multiplexing up to four RNA targets, simultaneously, and (v) robustness that ease brain mapping [21]. In this study, we successfully combined the RNAscope technology with sequential indirect immunofluorescence for commonly used and well-trusted neurochemical markers.

RNAscope-based *Sstr4* transcript expression was strongly supported by the conventional indirect β-Gal IHC using the *Sstr4^lacZ/lacZ^* reporter strain and by the RT-qPCR results. Although the main problem of the β-Gal IHC is that co-localization studies with specific markers of distinct neuronal populations cannot be performed due to need for different technical conditions, it is an important method to confirm *Sstr4* expression pattern at the mRNA level. 

In addition, our electrophysiological experiments showed that SST_4_ activation by the highly selective agonist J-2156 [26] decreased the excitability of layer V glutamatergic neurons. It had a similar tendency on layer II-IV glutamatergic neurons, but it had no effect on GABAergic interneurons. These data are fully supported by the abundant expression of SST_4_ in layer V pyramidal cells, low copy expression in layer II-IV pyramidal cells, but lack of expression in interneurons. Importantly, J-2156 was completely ineffective in case of layer V S1 glutamatergic neurons recorded from *Sstr4^lacZ/lacZ^* mice. Finally, we show for the first time, that SST_4_ activation augments M-current in neocortical layer V S1 glutamatergic neurons responsible for processing sensory stimuli, which is in parallel with earlier data from CA1 pyramidal cells [10].

A limitation of this study is that the mRNA values obtained from human and mouse samples of respective brain regions cannot be directly compared due to differences between their structures, functions and post-mortem durations. As intact primary somatosensory areas, nociceptive and anti-nociceptive brain centers are not rested in glioblastoma surgery, we could not show human RNAscope data in S1 cortex. Since the access to intact human cortex is very limited, we managed to collect oriented blocks of the left middle temporal gyrus obtained from neurosurgical interventions, processed specifically for this purpose and demonstrated *SSTR4* in glutamatergic neurons. Although this area functionally differs both from S1 and the limbic areas examined in the mouse, its involvement in psychosocial functions and mood regulation [44] further supports the translational value of our findings even with these limitations.

These first morphological and functional data are particularly valuable to understand the mechanism of action of SST_4_ agonists as novel analgesic drug candidates with simultaneous antidepressant effect. This unique dual indication is especially important, since chronic neuropathic pain, which is a huge unmet medical need, commonly occurs along with depression, worsening the disease. Therefore, SST_4_ activation as a new therapeutic approach seems to be a promising double hit as recognized by several pharma companies [25,45].

## 4. Materials and Methods

### 4.1. G-Protein Activation Assay

Membrane fractions (10 μg of protein/sample) were prepared from CHOcells stably expressing human SST_4_ receptor (prepared in our laboratory using lentiviral expression system) in Tris–EGTA buffer (50 mM Tris–HCl, 1 mM EGTA, 3 mM MgCl_2_, 100 mM NaCl, pH 7.4). These fractions were incubated at 30 °C for 60 min in the buffer containing 0.05 nM guanosine triphosphate, [^35^S]GTPγS (Institute of Isotopes, Budapest, Hungary) and 30 μM GDP (Merck KGaA, Darmstadt, Germany). Increasing concentrations (1 nM–10 µM) of J-2156 (Juvantia Pharma, Turku, Finland) was added in a final volume of 500 µL. The samples were filtrated by Whatman GF/B glass fiber filters using 48-well Slot Blot Manifold from Cleaver Scientific, filters were washed with ice-cold 50 mM Tris–HCl buffer (pH 7.4). Radioactivity was measured in scintillation liquid in a γ-counter after drying for 60 min at 37 °C. Total binding was determined in the absence of test compounds, and non-specific binding in the presence of 10 μM unlabeled GTPγS. Non-specific binding was subtracted from total binding to calculate the specific binding. J-2156-induced G-protein stimulation was given as percentage over the specific [^35^S]GTPγS binding observed in the absence of agonist [9].

### 4.2. Samples

#### 4.2.1. Mice

All animals were bred and kept in the Animal House of the Department of Pharmacology and Pharmacotherapy of the University of Pécs according to the regulations of the European Community Council Directive and the Animal Welfare Committee of the University of Pécs. 

#### 4.2.2. Human Post-Mortem and Neurosurgical Cortical Tissues

Various cortical and hippocampal tissues for the RT-qPCR studies were obtained from 4–8 patients without any major brain pathologies 1 to 10 h after death. Tissue collection was approved by the Medical Research Council (ETT TUKEB); approval number: 5912-2/2018/EKU (Human Brain Tissue Bank, Semmelweis University, Budapest, Hungary). For medical history of human patients, see Appendix A. For the RNAscope ISH, two neurosurgical samples were obtained from the intact left middle temporal gyrus of a 56-year-old man and a 19-year-old woman both operated due to glioblastoma multiforme. (Ethical approval number: 2446-2/2016/EKU). 

### 4.3. Sample Preparation and Investigational Techniques

#### 4.3.1. Mouse

##### Real-Time Quantitative RT-qPCR

Brains of 3–4 month-old male C57BL6/J (WT) mice were quickly dissected after decapitation and snap-frozen on dry ice. After storage at –80 °C, brains were sliced using razor blades on a coronal brain matrix (Ted Pella, Redding, CA, USA) to obtain 1 mm thick coronal sections, based on the technique of Palkovits et al. [46]. A microdissecting tool (Ted Pella, Redding, CA, USA) of 1 mm diameter was used to punch the brain areas of interest located in the following coronal sections (the distances of coronal sections from the Bregma were indicated in brackets, based on Paxinos and Franklin: prelimbic cortex (+2 mm–(+1 mm)), primary somatosensory cortex, hippocampus, habenula, piriform cortex, amygdala (−1 mm–(−2 mm)) [47]. The microdissection procedure was performed on a dry ice-chilled mat and the punches were immediately snap-frozen in pre-cooled Eppendorf vials on dry ice. Total RNA from microdissected mouse brain samples were extracted with Direct-zol RNA Microprep kit (Zymo Research, Irvine, CA, USA). 1 μg DNase 1-treated (Zymo Research) RNA was reverse transcribed into cDNA using Maxima First Strand cDNA Synthesis Kit (Thermo Fisher Scientific, Waltham, MA, USA). Stratagene Mx3000P QPCR System (Agilent Technologies, Santa Clara, CA, USA) was used to perform qPCR experiments using Luminaris Color HiGreen Low ROX qPCR Master Mix (Thermo Fisher Scientific) according to the manual. The qPCR reaction was run at the following program: 95 °C for 10 min, followed by 40 cycles of 95 °C for 30 s, 60 °C for 30 s and 72 °C for 45 s. All qPCR experiments were performed in technical replicates and included a melt curve analysis to ensure specificity of signal. Reverse transcriptase minus control showed lack of genomic DNA contamination. The geometric mean of the reference gene Ct values was determined and *Sstr4* mRNA expression relative to the reference genes (β-glucuronidase (*Gus*) and hydroxymethylbilane synthase (*Hmbs*)) was calculated using the 2^−ΔCt^ formula to compare different brain regions [48]. All primers are listed in Appendix A.

##### Perfusion and Tissue Processing for Histological Studies

For all histological studies (β-galactosidase, SST_4_ immunohistochemistry and RNAscope) were performed on 3–4 month-old male WT or *Sstr4^lacZ/lacZ^* mice. Animals were deeply anesthetized with an overdose of urethan (2.4 g/kg) and perfused transcardially with 30 mL of 4% paraformaldehyde in Millonig’s phosphate buffer. For β-gal and SST_4_ IHC dissected brains were postfixed for 72 h at room temperature (RT), and sectioned (by 30 µm) using a vibrating microtome (VT1000S, Leica Biosystems, Germany). For RNAscope brains, spinal cords and trigeminal ganglia (TG) were postfixed for 24 h at RT, rinsed in 1× PBS, dehydrated, and embedded in paraffin using standard procedures. The 5 µm sections were cut using a sliding microtome (HM 430, Thermo Fisher Scientific, USA); L5 dorsal root ganglia (DRG) were postfixed for 24 h at 4 °C, cryoprotected in 30% sucrose in 1× PBS for 24 h at 4 °C and frozen in tissue freezing media (Leica Biosystems, Wetzlar, Germany) on dry ice. Moreover, 20 µm sections were cut using cryostat (CM1850, Leica Biosystems).

##### β-Galactosidase-Specific Immunohistochemistry (β-Gal IHC)

Sections were washed 3 times in 1× TBS and the endogenous peroxidase activity was blocked with 0.3% H_2_O_2_. Blocking buffer contained 2% bovine serum albumin and 0.3% Triton X-100 (both purchased from Merck KGaA). Then sections were incubated overnight at RT with polyclonal chicken anti-β-galactosidase (ab9361, RRID: AB_307210, Abcam, Cambridge, UK), diluted 1:4000 in blocking buffer. After TBS washes, sections were incubated for 2 h at RT with peroxidase conjugated donkey anti-chicken IgG (Jackson ImmunoResearch Laboratories Inc., West Grove, PA, USA), diluted 1:500 in blocking buffer. Visualization were performed using nickel(II) sulfate hexahydrate/3,3′-diaminobenzidine tetrahydrochloride (Merck KGaA) as chromogen and glucose oxidase (Merck KGaA) [49]. Sections were mounted onto gelatinized slides, allowed to dry overnight, transferred into ascending ethanol solutions (50%, 70%, 96%, absolute for 5 min, respectively), then cover slipped with PERTEX mounting medium. Brightfield images of PrL, BMA, CeC, BLA, layer V of the primary S1, CA1 and MHb, according to Paxinos and Franklin [47] were acquired using a Nikon Microphot-FXA microscope (Nikon, Tokyo, Japan), then contrasted using Photoshop CS6 (Adobe, San José, CA, USA). 

##### SST_4_-Specific Immunohistochemistry

Sections were washed 3 times in 1× PBS and the endogenous peroxidase activity was blocked with 0.3% H_2_O_2_. Antigens were retrieved by using sodium citrate buffer (10 mM Sodium Citrate, 0.05% Tween 20, pH 6.0) for 10 min at 90 °C and permeabilized with 0.5% Triton X-100. After blocking (30 min with 2% NGS), sections were incubated with anti-SST_4_ antibodies (PA3-208 from Thermo Fisher Scientific diluted to 1:500 and 1:2000 in 2% NGS; GTX70677 from GeneTex, Irvine, CA, USA diluted to 1:100 and 1:200 in 2% NGS) for overnight at RT. Fluorescent immunostaining was performed using goat anti-rabbit IgG (H+L) cross-adsorbed secondary antibody, Alexa Fluor 594 (A-11012, Thermo Fisher Scientific) diluted to 1:1000 in 1× PBS for 3 h at RT. Sections were washed and mounted on adhesive slides with VECTASHIELD HardSet Antifade Mounting Medium with DAPI (H-1500, Vector Laboratories, Burlingame, CA, USA). For IHC by DAB, sections were treated with 1:200 diluted biotinylated goat anti-rabbit antibody for 60 min (Vectastain ABC Elite Kit, Vector Laboratories), then with peroxidase conjugated avidin–biotin complex (Vectastain ABC Elite Kit, Vector Laboratories) according to the supplier’s protocol for 60 min. After rinses in PBS, the immunoreaction was developed in 1× TBS (pH 7.4) with 0.02% 3,3′ DAB (Merck KGaA) and 0.03 *v/v*% H_2_O_2_. The chemical reaction was carried out under visual control using a microscope to optimize the signal/background ratio, and was stopped after 10 min with 1× PBS. Then, preparations were washed with 1× PBS and mounted on gelatin-covered slides. After drying, slides were transferred into ascending ethanol solutions (50%, 70%, 96%, absolute for 5 min, respectively), then into xylene for 2 × 10 min and cover slipped using DEPEX (VWR, West Chester, PA, USA) mounting medium. Brightfield and fluorescent images of CA1 and dentate gyrus (DG), according to Paxinos and Franklin [47] were acquired using a Nikon Eclipse Ti2 microscope (Nikon), then contrasted using Photoshop CS6 (Adobe).

##### RNAscope In Situ Hybridization (ISH) on Mouse Samples

RNAscope assay was performed on 5 µm thick formalin-fixed paraffin-embedded coronal brain, TG and spinal cord sections, and 20 µm thick fixed frozen DRG sections using RNAscope Multiplex Fluorescent Reagent Kit v. 2 (Advanced Cell Diagnostics, Newark, CA, USA) according to the manufacturer’s protocol. Briefly, after tissue pretreatment, sections were hybridized with probes specific to mouse *Sstr4*, *Vglut1*, (vesicular glutamate transporter 1), *Chat* (choline acetyltransferase), *Gad1* (glutamate decarboxylase 1) and *Rbfox3* (RNA Binding Fox-1 Homolog 3, NeuN) mRNA. Signal amplification and channel development were applied sequentially. Sections were counterstained with 4′,6-diamidino-2-phenylindole (DAPI) and mounted with ProLong Diamond Antifade Mountant (Thermo Fisher Scientific) for confocal imaging. Probes, applied dilutions of fluorophores are listed in Appendix A. RNAscope 3-plex mouse positive and negative control probes were used in parallel to ensure interpretable results. Fluorescent images of the DRG, spinal cord, TG, PrL, MPO, BMA, CeC, BLA, S1, CA1, MHb, VPL, VLPAG, RMg and LC according to Paxinos and Franklin [47], were acquired using an Olympus Fluoview FV-1000 laser scanning confocal microscope and FluoView FV-1000S-IX81 image acquisition software system (Olympus, Tokyo, Japan). The confocal aperture was set to 80 µm. The analog sequential scanning was performed using a 40× objective lens (NA: 0.75). The optical thickness was set to 1 μm and resolution was 1024 × 1024 pixel. The excitation time was set to 4 µs per pixel. Virtual colors were selected to depict fluorescent signals: blue for DAPI, green for fluorescein (*Vglut1* and *Chat* mRNA), red for Cyanine 3 (*Sstr4* mRNA), and white for Cyanine 5 (*Gad1* and *Rbfox3* mRNA). Images of the respective four channels were stored both individually, and superimposed to evaluate the co-localization of fluorescent signals. The RNAscope ISH signal for *Sstr4* mRNA was combined with immunofluorescence in the RMg for serotonin [monoclonal mouse anti-serotonin serum ((1:20,000); gift from Dr. Lucienne Léger, Université Claude Bernard, Lyon, France [50,51]). To assess the possible co-localization of *Sstr4* mRNA with the norepinephrinergic cells of the LC, tyrosine-hydroxylase (TH, polyclonal rabbit anti-tyrosine-hydroxylase, ab6112, 1:2000, RRID: AB_297840, Abcam) labeling was also performed. After the RNAscope ISH, the cocktail of primary antisera was applied overnight at RT. After rinses, Alexa Fluor 488 conjugated donkey anti-mouse (715-545-150, Jackson ImmunoResearch Europe Ltd., Cambridge, UK) and Alexa Fluor 647 conjugated donkey anti rabbit (711-605-152, Jackson ImmunoResearch Europe Ltd.) sera (1:500) were used as secondary antibodies for 3 h at RT. After washes, sections were counterstained with DAPI, coverslipped and examined as described above for RNAscope ISH. Green and white virtual colors were assigned to Alexa Fluor 488 and 647, respectively.

##### Acute Brain Slice Preparation

Electrophysiology experiments were performed in acute coronal slices taken from 30–35-day-old male WT or *Sstr4^lacZ/lacZ^* mice. Under deep isoflurane anesthesia, mice were decapitated and 350 μm thick coronal slices were cut in ice-cold external solution containing (in mM): 93 NMDG, 2.5 KCl, 25 Glucose, 20 HEPES, 1.2 NaH_2_PO_4_, 10 MgSO_4_, 0.5 CaCl_2_, 30 NaHCO_3_, 5 L-ascorbate, 3 Na-pyruvate, 2 thiourea bubbled with 95% O_2_ and 5% CO_2_. Slices were transferred to artificial cerebrospinal fluid containing (in mM) 2.5 KCl, 10 glucose, 126 NaCl, 1.25 NaH_2_PO_4_, 2 MgCl_2_, 2 CaCl_2_, 26 NaHCO_3_ bubbled with 95% O_2_ and 5% CO_2_. After an incubation period of 10 min at 34 °C in the first solution, the slices were maintained at 20–22 °C in ACSF until use. After recordings, the sections were immersed into fixative (4% paraformaldehyde with 0.1% picric acid in 0.01 M PB) for overnight fixation.

##### In Vitro Electrophysiological Recordings

Patch pipettes were pulled from borosilicate glass capillaries with filament (1.5 mm outer diameter and 1.1 mm inner diameter; Sutter Instruments, Novato, CA, USA) with a resistance of 2–3 MΩ. The pipette recording solution contained (in mM) 5 KCl, 135 K-gluconate, 1.8 NaCl, 0.2 EGTA, 10 HEPES, 2 Na-ATP and 0.2% Biocytin, pH 7.3 adjusted with KOH; 290–300 mOsm. Whole-cell recordings were made with Axopatch 700B amplifier (Molecular Devices, San José, CA, USA) using an upright microscope (Eclipse FN1, Nikon) with 40× (NA: 0.8) water immersion objective lens equipped with differential interference contrast (DIC) optics. DIC images were captured with an Andor Zyla 5.5 s CMOS camera (Oxford Instruments, Abingdon, UK). All recordings were performed at 32 °C, in ACSF bubbled with 95% O_2_ and 5% CO_2_. Cells with lower than 20 MΩ access resistance (continuously monitored) were accepted for analysis. Signals were low-pass filtered at 5 kHz and digitized at 20 kHz (Digidata 1550B, Molecular Devices). When it is indicated 1 µM J-2156 was applied to the bath solution. For the M-current recordings cells were washed with 1 µM TTX (Merck KGaA) and held on −0 mV holding potential and 1 s long repolarization step to −0 mV was applied. 

#### 4.3.2. Human

##### RT-qPCR on Post-Mortem Human Cortical Tissues

Brain samples were quickly homogenized using a dispenser (T 25 digital Ultra-TURRAX, IKA-Werke, Staufen, Germany). Total RNA were extracted, purity control and generation of cDNA pool were already above described. The samples were stored in airtight containers or plastic tubes at −80 °C until further use. Stratagene Mx3000P QPCR System was used to perform qPCR experiments using SensiFAST SYBR Lo-ROX Kit (Meridian Bioscience, Cincinnati, OH, USA) according to the manual. The qPCR reaction mixture was run in a 20 μL reaction volume (20 ng of cDNA, 1× qPCR Master Mix, *POLR2A* forward primer (F): 0.15 µM, reverse primer (R): 0.15 µM, probe (P): 0.2 µM; *PES1* F: 0.4 µM, R: 0.2 µM, P: 0.4 µM; *IPO8* F: 0.4 µM, R: 0.4 µM, P: 0.4 µM; *SSTR4* F: 0.3 µM; R: 0.6 µM; P: 0.3 µM) using the following program: 9 °C for 2 min, followed by 40 cycles of 95 °C for 10 s and 60 °C for 30 s. All qPCR experiments were performed in technical replicates. The geometric mean of the reference gene Ct values was determined and *SSTR4* mRNA expression relative to the reference genes (DNA-directed RNA polymerase II subunit RPB1 (*POLR2A*), pescadillo homolog (*PES1*), importin 8 (*IPO8*)) was calculated using the 2^−ΔCt^ formula to compare different brain regions. All primers are listed in Appendix A.

##### RNAscope ISH on Neurosurgical Cortical Tissue Samples

Sample A was freshly post-fixed for 24 h at RT in 10% neutral buffered formalin (Merck KGaA) after surgery, while Sample B was snap-frozen in liquid N_2_ and stored at –80 °C until further use. Upon thawing, Sample B was postfixed as described above. Then both samples were rinsed in 1× PBS, dehydrated, and embedded in paraffin using standard procedures. 5 µm sections were cut using a sliding microtome. RNAscope assay was performed on 5 µm thick sections using RNAscope Multiplex Fluorescent Reagent Kit v. 2 (Advanced Cell Diagnostics) according to the manufacturer’s protocol hybridizing with human *SSTR4* (red) and *VGLUT1* (green) probes. RNAscope 3-plex human positive and negative control probes were used in parallel to ensure interpretable results. Probes, applied dilutions of fluorophores are listed in Appendix A.

### 4.4. Experimental Design and Statistical Analysis

For G-protein activation assay membrane fractions prepared from CHO cells stably expressing human SST_4_ receptor were used. For all histological and RT-qPCR studies 3–4 month-old C57BL6/J (WT) or *Sstr4^lacZ/lacZ^* mice were used. For electrophysiological experiments 30–35-day-old male C57BL6/J (WT) or *Sstr4^lacZ/lacZ^* mice were used. Human post-mortem brain samples without any major brain pathologies were collected for RT-qPCR, and neurosurgical cortical tissues obtained from the intact left middle temporal gyrus of patients operated due to glioblastoma multiforme were collected for RNAscope ISH. Animal/human patient (*N*) and cell/experiment (*n*) numbers are reported in the figure legends and in the result section. All statistical analyses of electrophysiological data were performed using Clampfit v. 10.7 (Molecular Devices) and OriginPro v. 8.6. Statistical analysis was performed by paired Student’s *t*-test.

## Figures and Tables

**Figure 1 ijms-21-07788-f001:**
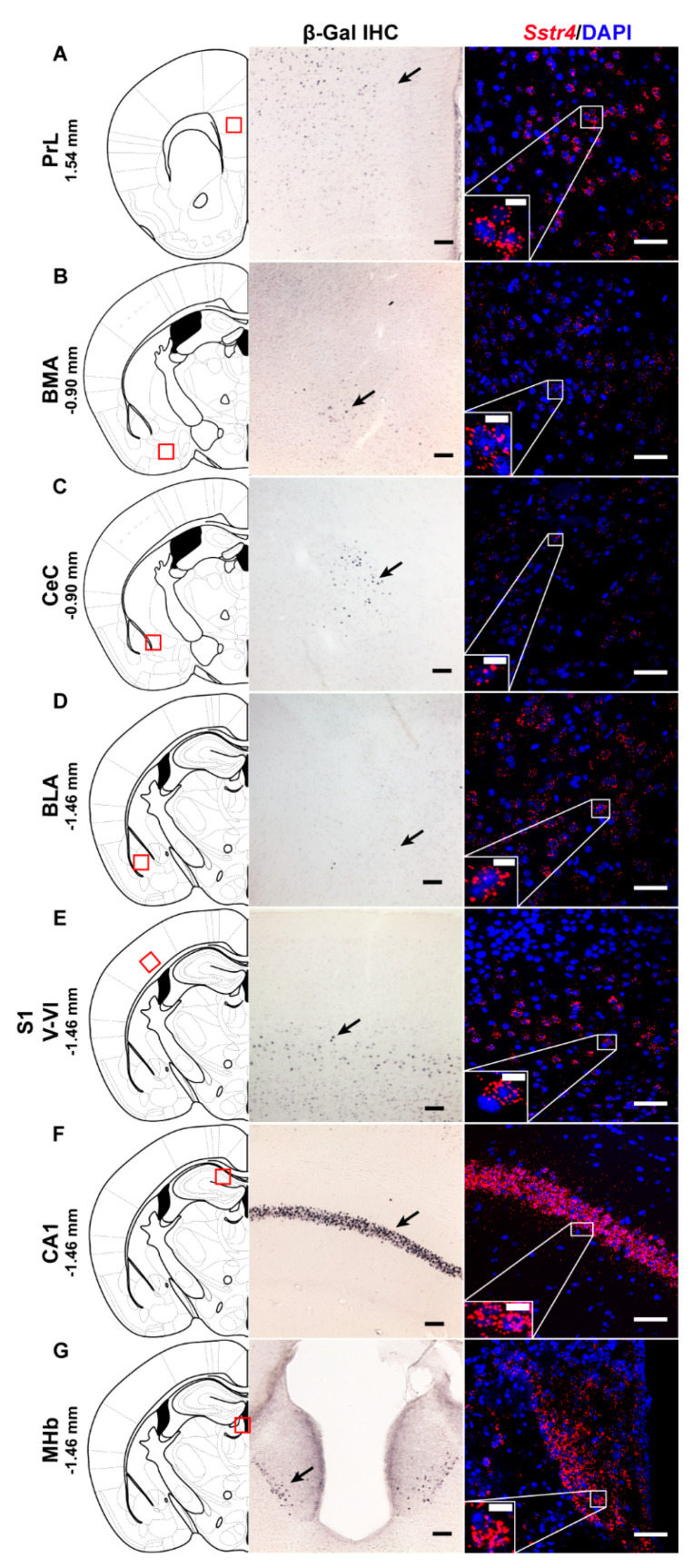
Representative β-galactosidase and *Sstr4* mRNA expression in various mouse brain areas (by β-Gal immunohistochemistry and singleplex fluorescent RNAscope). *lacZ* signal from *Sstr4^lacZ/lacZ^* mice (*N* = 3, brightfield, left panels) and *Sstr4* mRNA in red counterstained with DAPI from C57BL6/J (WT) mice (*N* = 4, right panels) are shown in the prelimbic cortex (PrL, Bregma 1.54 mm, (**A**)), in the basomedial amygdala (BMA, Bregma −0.9 mm, (**B**)), in the core of central amygdala (CeC, Bregma −0.9 mm, (**C**)), in the basolateral amygdala (BLA, Bregma −1.46 mm, (**D**)), in the layer V pyramidal neurons of the primary somatosensory cortex (S1, Bregma −1.46 mm, (**E**)), in the CA1 of hippocampus (CA1, Bregma −1.46 mm, (**F**)), and in the medial nucleus of habenula (MHb, Bregma −1.46 mm, (**G**)). Scale bar: 50 µm, inset scale bar: 10 µm.

**Figure 2 ijms-21-07788-f002:**
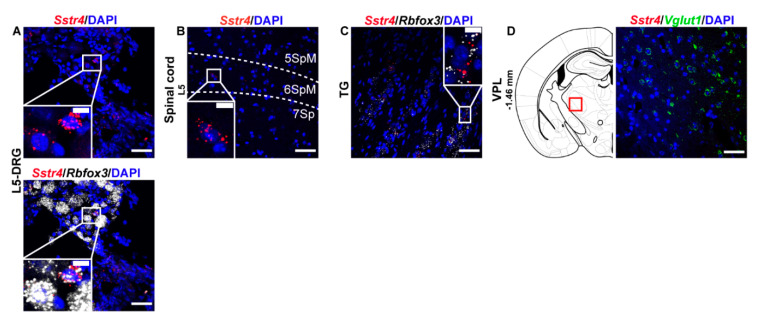
Representative *Sstr4* mRNA expression counterstained with DAPI in the primary nociceptive spinothalamic system in WT mice. *Sstr4* mRNA (red, upper and lower panel) co-localized with *Rbfox3* mRNA (NeuN, white, lower panel) and are shown in the L5 dorsal root ganglion (DRG) (**A**). *Sstr4* mRNA (red) is shown in the in the laminae V-VII of the spinal cord of L4–L6 ((**B**)), in the trigeminal ganglion (TG, (**C**)) co-localized with *Rbfox3* mRNA (NeuN, white), and in the ventral posterolateral thalamic nucleus (VPL, Bregma −1.46 mm, (**D**)) co-localized with *Vglut1* (green) mRNA. Note that *Sstr4* cannot be detected in VPL area. *N* = 3. Scale bar 50 µm, inset scale bar: 10 µm.

**Figure 3 ijms-21-07788-f003:**
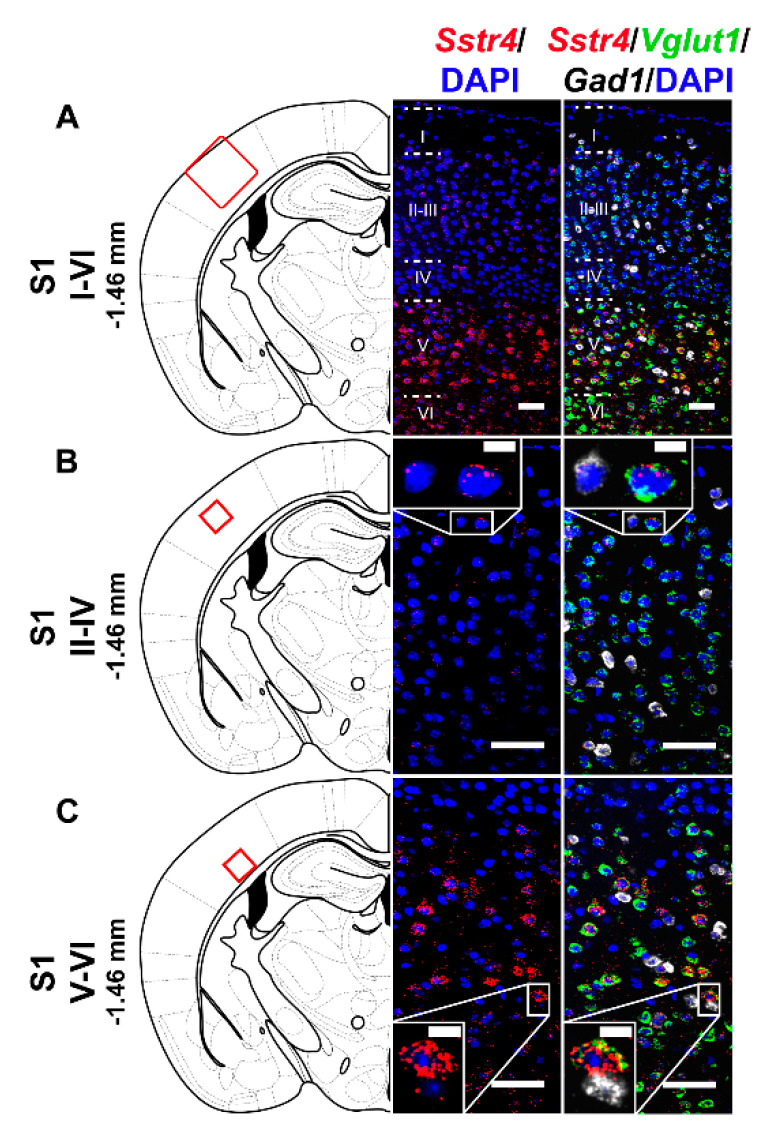
Representative *Sstr4*, *Vglut1* and *Gad1* mRNA expression in the primary somatosensory cortex (S1) of WT mice. *Sstr4* (red), *Vglut1* (green), and *Gad1* mRNA (white) expression counterstained with DAPI are shown in the S1 I–VI (Bregma −1.46 mm, (**A**)), in the S1 II–IV (**B**) and in the S1 V–VI (**C**) (*N* = 5). Scale bar: 50 µm, inset scale bar: 10 µm.

**Figure 4 ijms-21-07788-f004:**
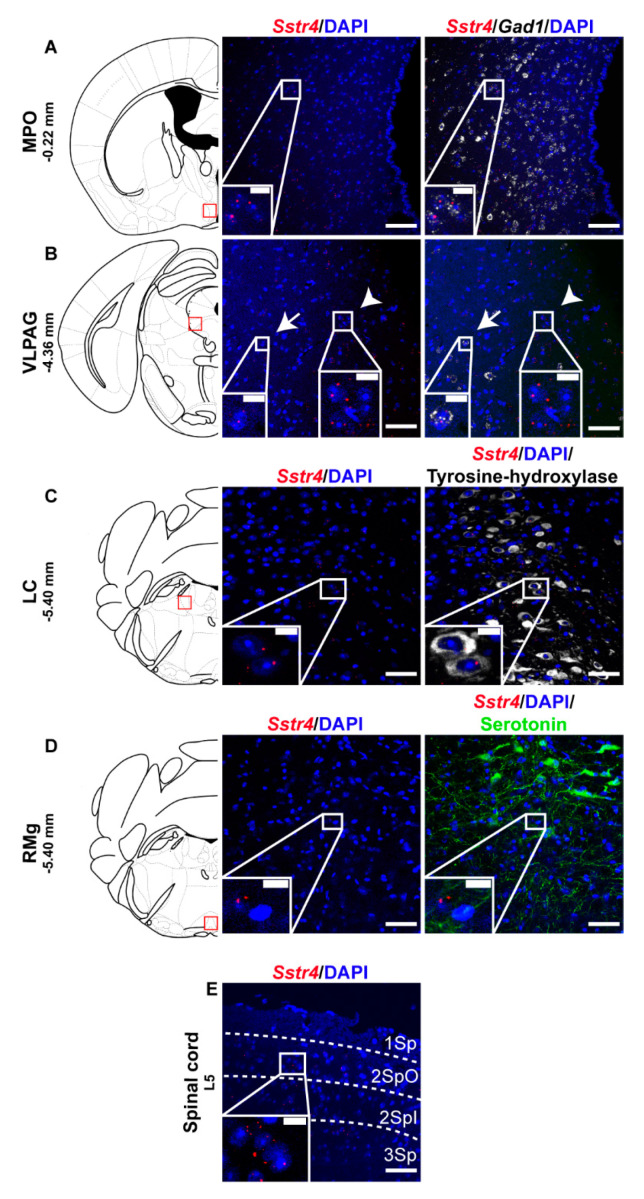
Representative *Sstr4* expression in the descending anti-nociceptive system of WT mice. *Sstr4* mRNA (red) counterstained with DAPI are shown in the medial preoptic area (MPO, Bregma −0.22 mm, (**A**)), in the ventrolateral periaqueductal gray matter (VLPAG, Bregma −4.36 mm, (**B**), arrow marks *Sstr4* co-localization with *Gad1*, arrowhead points to *Gad1*-negative *Sstr4* expressing neurons), in the locus coeruleus (LC, Bregma −5.40 mm, (**C**)) co-localized with tyrosine-hydroxylase (TH)-immunoreactive noradrenergic cells (white) and in the raphe magnus nucleus (RMg, Bregma −5.40 mm, (**D**)) where *Sstr4* does not co-localize with serotoninergic cells (green). Panel (**E**) represents *Sstr4* mRNA expressing cells in the superficial layers (I–III) of the spinal dorsal horn. (*N* = 2). Scale bar: 50 µm, inset scale bar: 10 µm.

**Figure 5 ijms-21-07788-f005:**
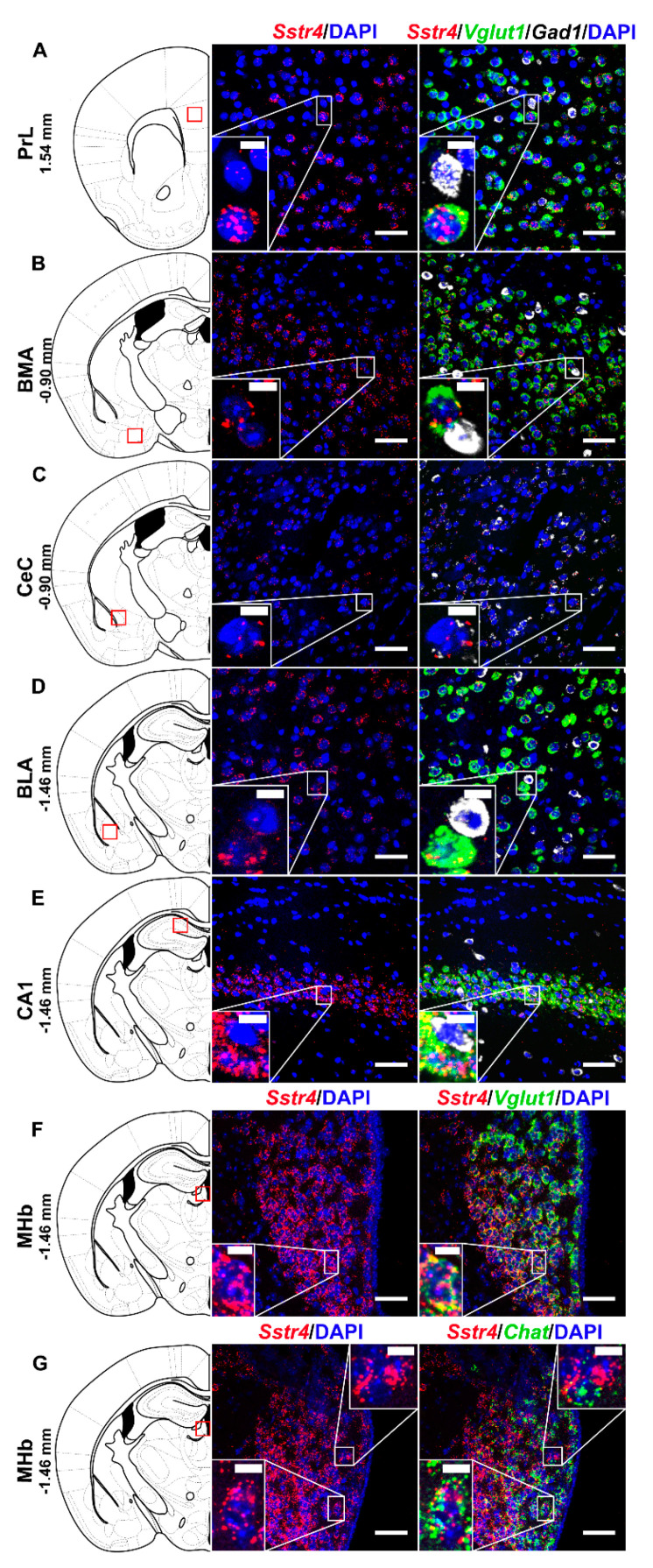
Representative *Sstr4*, *Vglut1*, *Gad1 and Chat* mRNA expression in various brain areas of WT mice. *Sstr4* (red), *Vglut1* (green), and *Gad1* mRNA (white) expression counterstained with DAPI are shown in the prelimbic cortex (PrL, Bregma 1.54 mm, (**A**)), in the basomedial amygdala (BMA, Bregma −0.9 mm, (**B**)), in the core of central amygdala (CeC, Bregma −0.9 mm, (**C**)), in the basolateral amygdala (BLA, Bregma −1.46 mm, (**D**)), in the CA1 of hippocampus (CA1, Bregma −1.46 mm, (**E**)) (*N* = 5). *Sstr4* (red) and *Vglut1* (green) expression counterstained with DAPI are shown in the medial nucleus of habenula (MHb, Bregma −1.46 mm, *N* = 3, (**F**)). *Sstr4* (red) and *Chat* (green) expression counterstained with DAPI are shown in the MHb (*N* = 2, (**G**)). Scale bar: 50 µm, inset scale bar: 10 µm.

**Figure 6 ijms-21-07788-f006:**
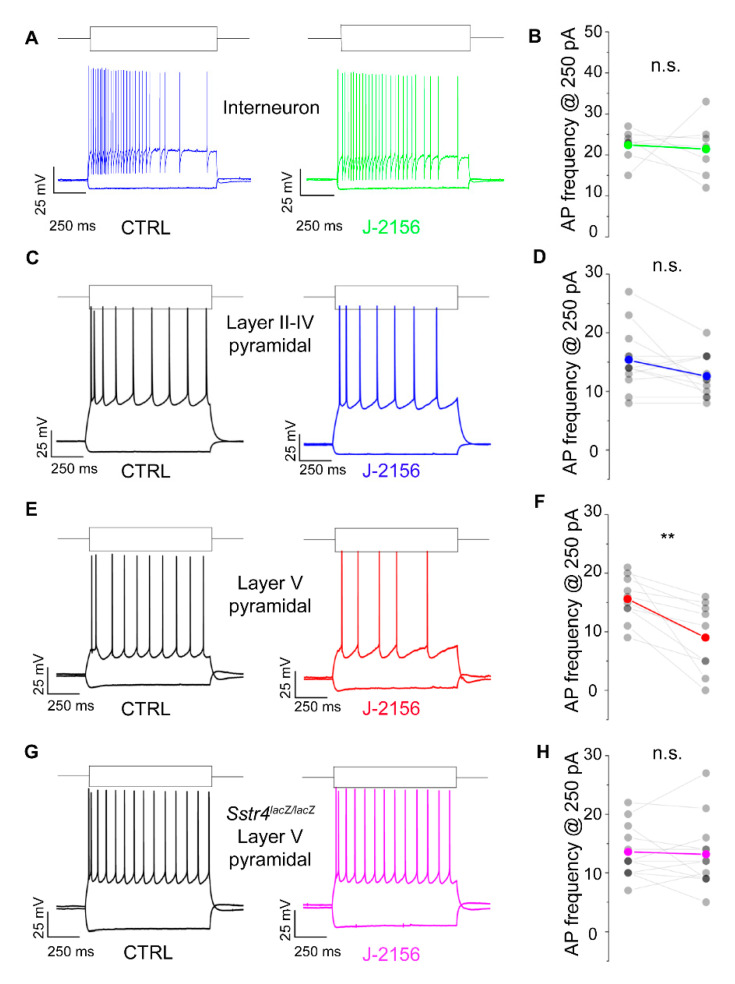
SST_4_ receptor activation decreases firing frequency of layer V pyramidal cells of the primary somatosensory cortex (S1). Representative voltage responses of interneurons (*n* = 7, (**A**)), layer II–IV pyramidal cells (*n* = 12, (**C**)), and layer V pyramidal cells (*n* = 10, (**E**)) of the somatosensory cortex upon step current injections (250 and –200 pA) under control conditions (black) and after bath application of 1 µM J-2156 (colored; (**B**,**D**,**F**,**H**)). Layer V pyramidal cells of *Sstr4^lacZ/lacZ^* mice not showing any response to J-2156 served as negative control (*n* = 12, (**G**)). Statistical analysis was performed by paired Student’s *t*-test on the firing frequency of corresponding cells upon 250 pA step current injections under control and J-2156 treated conditions. Paired Student’s *t*-test **: *p* = 0.003.

**Figure 7 ijms-21-07788-f007:**
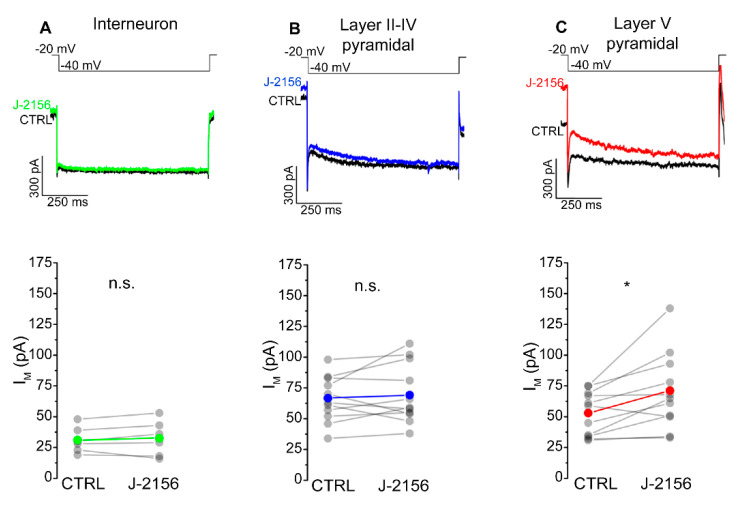
SST_4_ receptor activation augments M-current in layer V pyramidal cells of the primary somatosensory cortex (S1). Representative current traces from interneurons (*n* = 6, (**A**)), somatosensory cortex layer II–IV pyramidal cells (*n* = 11, (**B**)), layer V pyramidal cells (*n* = 11, (**C**)), and elicited by the voltage protocol indicated at the top of each panel under control conditions (black) and after bath application of 1 µM J-2156 (colored). M-current was determined as the difference of the instantaneous current at the beginning of the current trace, and the steady current at the end of the current trace. Bottom side: statistics showing M-current amplitude of corresponding cells under control and J-2156 treated conditions. Paired Student’s *t*-test *: *p* = 0.016.

**Figure 8 ijms-21-07788-f008:**
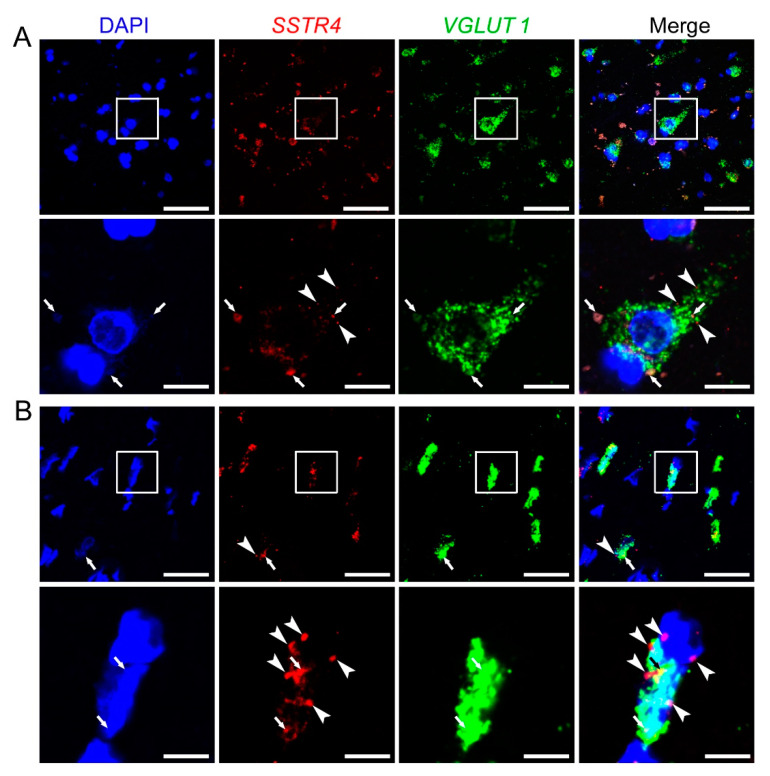
Representative *SSTR4* and *VGLUT1* mRNA expression of layer V pyramidal neurons of human neurosurgical cortical samples taken from the left middle temporal gyrus (by multiplex fluorescent RNAscope in situ hybridization (ISH)). Panel (**A**) shows images from Sample A of a 56-year-old man. Note that the tissue contains considerable amount of lipofuscin that shows some autofluorescence in all examined channels (arrows). Granular fluorescence pattern that occurs in one color only represents the respective specific RNAscope ISH signal for *VGLUT1* (green) or *SSTR4* (red, marked in part by arrowheads) counterstained with DAPI (blue). Panel (**B**) depicts images of Sample B taken from a 19-year-old woman. Note the relatively low aging pigment content (arrows) of the *VGLUT1* (green) expressing layer V pyramidal cells, which contain *SSTR4* mRNA (arrowheads) also, counterstained with DAPI (blue). Cellular and nuclear (DAPI) morphology in Sample B differs from that of Sample A, as Sample B was fresh frozen before fixation, in contrast to Sample A that was subjected to immediate immersion fixation in 10% neutral buffered formalin. Boxed areas in the top images (scale bar: 100 µm) of both panels are shown in higher magnification photos right bellow (scale bar: 25 µm).

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
