# Peer review of "Characterization of Neurons Expressing the Novel Analgesic Drug Target Somatostatin Receptor 4 in Mouse and Human Brains"

_ijms, 2020, doi:10.3390/ijms21207788_

Round 1
Reviewer 1 Report
As fas as I can tell the authors have shown no robust proof that SST4 is specific for neuropathic pain and neither is there any robust evidence that SST4 is of any use for the treatment of depression. These are assumptions and speculations from the mouse data. The significance of the human post mortem data is not clear.
Author Response
Response to Reviewer 1:
We thank the review for this comments and we hope that our answers will be satisfactory.
“As fas as I can tell the authors have shown no robust proof that SST4 is specific for neuropathic pain and neither is there any robust evidence that SST4 is of any use for the treatment of depression. These are assumptions and speculations from the mouse data. The significance of the human post mortem data is not clear.”
We did not state that the “SST4 is specific for neuropathic pain”, this is, of course, not the message and conclusion of our paper. The main focus of our study was to provide molecular mechanism of action to explain the well-established functional results obtained with SST4 KO animals [1–3] and SST4 agonists [4–8].
These already published results of ours and other groups provided strong proof-of-concept for SST4 to become and interesting and novel analgesic drug target. SST4 agonists are currently going through preclinical development by big pharma [9] and SMEs [10]. Therefore, it is important to explore the expression profile and understand the pharmacodynamic characteristics. We strongly believe that the present data provide useful and novel information to this field.
We did not write that the present data provide “any robust evidence that SST4 is of any use for the treatment of depression” besides the well-established analgesic drug developmental potential. Our conclusion regarding the potential anti-depressant role of the SST4 receptor is based on (i) the present expression profile in mood regulation-related brain regions that are mainly overlapping with areas involved in pain processing, as well as (ii) earlier functional data obtained with SST4 KO animals and an SST4 agonist [3]. Therefore, we concluded that: SST4 is a unique novel drug target for neuropathic pain, which could be also valuable to treat depression as a common comorbidity (lines 94-95).
We are aware that the translational relevance of the mouse data is limited, but rodent models are the only available and valuable tools in preclinical drug discovery and development. Investigating human tissue samples is indeed important but ethically, logistically and technically very challenging and difficult, particularly in case of the brain. There are several values to determine translational relevance of animal data, but also limitations for both post-mortem and neurosurgical samples. The advantage of post-mortem samples is that broad-range of regions are available from the Human Brain Bank, but the divergent post-mortem times and the other pathologies could be result-modifying factors. These frozen samples are optimal for molecular biological investigations of the homogenate e.g. RT-qPCR, but not for morphology/imaging (RNAscope). Therefore, for the latter purpose we used the neurosurgical samples, which can be processed immediately, and the morphological integrity can be preserved. However, these samples are only available from very limited regions
- Helyes, Z.; Pintér, E.; Sándor, K.; Elekes, K.; Bánvölgyi, Á.; Keszthelyi, D.; Szoke, É.; Tóth, D.M.; Sándor, Z.; Kereskai, L.; et al. Impaired defense mechanism against inflammation, hyperalgesia, and airway hyperreactivity in somatostatin 4 receptor gene-deleted mice. Proc. Natl. Acad. Sci. U. S. A. 2009, 106, 13088–13093, doi:10.1073/pnas.0900681106.
- Scheich, B.; Csekő, K.; Borbély, É.; Ábrahám, I.; Csernus, V.; Gaszner, B.; Helyes, Z. Higher susceptibility of somatostatin 4 receptor gene-deleted mice to chronic stress-induced behavioral and neuroendocrine alterations. Neuroscience 2017, 346, 320–336, doi:10.1016/j.neuroscience.2017.01.039.
- Scheich, B.; Gaszner, B.; Kormos, V.; László, K.; Ádori, C.; Borbély, É.; Hajna, Z.; Tékus, V.; Bölcskei, K.; Ábrahám, I.; et al. Somatostatin receptor subtype 4 activation is involved in anxiety and depression-like behavior in mouse models. Neuropharmacology 2016, 101, 204–215, doi:10.1016/j.neuropharm.2015.09.021.
- Helyes, Z.; Pintér, E.; Németh, J.; Sándor, K.; Elekes, K.; Szabó, Á.; Pozsgai, G.; Keszthelyi, D.; Kereskai, L.; Engström, M.; et al. Effects of the somatostatin receptor subtype 4 selective agonist J-2156 on sensory neuropeptide release and inflammatory reactions in rodents. Br. J. Pharmacol. 2006, 149, 405–15, doi:10.1038/sj.bjp.0706876.
- Kántás, B.; Börzsei, R.; Szőke, É.; Bánhegyi, P.; Horváth, Á.; Hunyady, Á.; Borbély, É.; Hetényi, C.; Pintér, E.; Helyes, Z. Novel Drug-Like Somatostatin Receptor 4 Agonists are Potential Analgesics for Neuropathic Pain. Int. J. Mol. Sci. 2019, 20, E6245, doi:10.3390/ijms20246245.
- Szőke, É.; Bálint, M.; Hetényi, C.; Markovics, A.; Elekes, K.; Pozsgai, G.; Szűts, T.; Kéri, G.; Őrfi, L.; Sándor, Z.; et al. Small molecule somatostatin receptor subtype 4 (sst4) agonists are novel anti-inflammatory and analgesic drug candidates. Neuropharmacology 2020, doi:10.1016/j.neuropharm.2020.108198.
- Markovics, A.; Szőke, É.; Sándor, K.; Börzsei, R.; Bagoly, T.; Kemény, Á.; Elekes, K.; Pintér, E.; Szolcsányi, J.; Helyes, Z. Comparison of the anti-inflammatory and anti-nociceptive effects of cortistatin-14 and somatostatin-14 in distinct in vitro and in vivo model systems. J. Mol. Neurosci. 2012, 46, 40–50, doi:10.1007/s12031-011-9577-4.
- Qiu, C.; Zeyda, T.; Johnson, B.; Hochgeschwender, U.; De Lecea, L.; Tallent, M.K. Somatostatin receptor subtype 4 couples to the M-current to regulate seizures. J. Neurosci. 2008, doi:10.1523/JNEUROSCI.4679-07.2008.
- https://www.lilly.com/discovery/clinical-development-pipeline#/
- https://algonist.com/services/
Reviewer 2 Report
Kecskés A and colleagues used sensitive RNAscope method to investigate Sstr4 expression in mouse and human brains. The authors revealed Sstr4 transcript expression pattern in the nuclei of the mouse nociceptive and anti-nociceptive pathways as well as in human brain regions. Moreover, using patch-clamp, the authors showed activation of the SST4 receptor by J-2156 decreased the excitability of layer V Pyramidal neurons. Although this is an interesting study and proposes new information on Sstr4 expression pattern in mouse and human brains, additional evidence from immunohistochemistry study is needed for robust conclusions.
Major concerns:
1. The authors claimed that specific antibodies for reliable determination of the SSTR4 are currently not available. However, several antibodies recognising rodent and /or human SSTR4 are commercially available: such as Catalog # PA3-208 (ThermoFisher), catalog No:orb79888 (Biorbyt), Cat# LS‑A4148 (LSBio) etc. The authors should also verify their expression findings at protein level rather than solely on the mRNA level.
2.Fig. 2 N numbers are too low. Experiment should be repeated at least by three times.
Author Response
Response to Reviewer 2:
We thank the review for this comments and we hope that our answers will be satisfactory.
“The authors claimed that specific antibodies for reliable determination of the SSTR4 are currently not available. However, several antibodies recognizing rodent and /or human SSTR4 are commercially available: such as Catalog # PA3-208 (ThermoFisher), catalog No:orb79888 (Biorbyt), Cat# LS‑A4148 (LSBio) etc. The authors should also verify their expression findings at protein level rather than solely on the mRNA level.”
The antibodies: No:orb79888 (Biorbyt, https://www.labome.com/product/Biorbyt/orb79888.html ), Cat# LS‑A4148 (LSBio, https://www.lsbio.com/pathplus-antibodies/pathplus-sstr4-antibody-n-terminus-ihc-ls-a4148/789731 ) are available only as human-reactive SST4 antibodies. Neither in silico mouse immunoreactivity nor the epitope sequence are mentioned on the websites. Therefore, we have not tested any of them on mouse tissues.
However, we performed both DAB-IHC and IF staining to test catalog # PA3-208 (ThermoFisher Scientific, TFS) and GTX70677 (GeneTex). Briefly, 30 um coronal sections (Bregma -1.3 - -1.5 mm) of 3 month-old male C57Bl6/J and Sstr4lacZ/lacZ (Sstr4 KO)mice (n=2-2) were sliced using a vibratome (Leica, Germany). Antigens were retrieved by using sodium citrate buffer (10mM Sodium Citrate, 0.05% Tween 20, pH 6.0) for 10 min at 90°C and permeabilized with 0.5% Triton X-100. After blocking (30 min with 2% NGS) sections were incubated with anti-SST4 antibodies (PA3-208 diluted to 1:500 and 1:2000 in 2% NGS, GTX70677 diluted to 1:100 and 1:200 in 2% NGS) for O/N at RT. Fluorescent immunostaining was performed using goat anti-rabbit IgG (H+L) cross-adsorbed secondary antibody, Alexa Fluor 594 (A-11012, ThermoFisher Scientific) diluted to 1:1000 in 1x PBS for 3 hr at RT. Sections were washed and mounted on adhesive slides with VECTASHIELD HardSet Antifade Mounting Medium with DAPI (H-1500). For IHC by DAB, sections were treated with 1:200 diluted biotinylated goat anti-rabbit antibody for 60 min (Vectastain ABC Elite Kit, Vector Lbs., Burlingame, CA, USA), then with peroxidase conjugated avidin–biotin complex (Vectastain ABC Elite Kit) according to the supplier’s protocol for 60 min. After rinses in PBS, the immunoreaction was developed in Tris buffer (pH 7.4) with 0.02% 3,3′ DAB (Sigma) and 0.03 v/v% H2O2. The chemical reaction was carried out under visual control using a microscope to optimize the signal/background ratio, and was stopped after 10 min with 1x PBS. Then, preparations were washed with 1x PBS and mounted on gelatin-covered slides. After drying, slides were transferred into ascending ethanol solutions (50%, 70%, 96%, absolute, absolute 5 min, respectively), then into xylene for 2 × 10 min and cover slipped using Depex (Fluka, Heidelberg, Germany) mounting medium.
See image in attached word document.
Representative images of IF and DAB-IHC staining of coronal sections from C57Bl6/J and Sstr4 KO mice (n=2). A-B, sections were incubated with 1:100 diluted GTX70667 anti-SST4 antibody. C-D, sections were incubated with 1:200 diluted GTX70667 anti-SST4 antibody. E-F, sections were incubated with 1:500 diluted PA3-208 anti-SST4 antibody. G-H, sections were incubated with 1:2000 diluted PA3-208 anti-SST4 antibody. I, sections were incubated with 1:1000 diluted goat anti-rabbit IgG secondary antibody conjugated to Alexa Fluor 594 (A-11012, ThermoFisher Scientific), no primary antibody control. J, sections were incubated with 1:200 diluted biotinylated goat anti-rabbit antibody, DAB signal was ABC method-amplified, no primary antibody control. Scale bar is 50 µm, same magnification was used on every image.
We conclude from the above described SST4 antibody validation, that neither # PA3-208 (ThermoFisher Scientific, TFS) nor GTX70677 (GeneTex) showed immunoreactivity to SST4-positive hippocampal neurons. These results were expected, as we wrote in the MS: “…specific antibodies for reliable determination of the SST4 protein are currently not available.” Schreff et al. performed SST4-IHC with a custom-made antibody that is not any more available either.
Our group has so far tested several other supernatants developed ImmunoGenes (Hungary) to generate custom-made monoclonal SST4 antibody. Until now, none of them were sensitive and specific enough in immunohistochemical/immunofluorescence conditions to give reliable immunosignal in the mouse brain none of them showed immunogenicity. We are currently developing more custom-made SST4 antibodies from ProteoGenix (France).
There are several reasons why SST4 is a notoriously difficult target to generate specific and effective antibodies against: (i) it is a GPCR and in general, GPCRs and other membrane proteins are difficult to express recombinantly and isolate in a soluble form to produce a native folded antigen; (ii) it belongs to a 5-member somatostatin receptor family, which show high identity at protein level; (iii) there are several PTMs on SST4 such as glycosylations and disulfide bonds that make even more challenging to find a specific epitope region.
Taken together, to overcome these difficulties, we chose RNAscope technology in parallel with SST4-augmented M-current detection using electrophysiology.
“Fig. 2 N numbers are too low. Experiment should be repeated at least by three times.”
We repeated this experiment, as the Reviewer suggested, and now we have n=3. This has been modified in the revised manuscript.

Round 2
Reviewer 2 Report
SST4 antibody validation results should be provided as a supplementary Fig. This will benefit other researchers.
Author Response
SST4 antibody validation results should be provided as a supplementary Fig. This will benefit other researchers.
Response to Reviewer 2:
We inserted the SST4 antibody validation results in the supplementary materials, as Fig. S3. See in the main text: line 144-146:
Additionally, we tested two commercially available antibodies (PA3-208 and GTX70677) on WT and Sstr4lacZ/lacZ mice. We could not detect any specific SST4 immunoreactivity either by immunofluorescence or 3,3′-diaminobenzidine (DAB) staining (Fig. S3).
Furthermore, method description see at M&M section, line: 477-498:
SST4-specific immunohistochemistry:
Sections were washed 3 times in 1x PBS and the endogenous peroxidase activity was blocked with 0.3% H2O2. Antigens were retrieved by using sodium citrate buffer (10mM Sodium Citrate, 0.05% Tween 20, pH 6.0) for 10 min at 90°C and permeabilized with 0.5% Triton X-100. After blocking (30 min with 2% NGS), sections were incubated with anti-SST4 antibodies (PA3-208 from Thermo Scientific, USA diluted to 1:500 and 1:2000 in 2% NGS; GTX70677 from GeneTex, USA diluted to 1:100 and 1:200 in 2% NGS) for overnight at RT. Fluorescent immunostaining was performed using goat anti-rabbit IgG (H+L) cross-adsorbed secondary antibody, Alexa Fluor 594 (A-11012, Thermo Scientific, USA) diluted to 1:1000 in 1x PBS for 3 hr at RT. Sections were washed and mounted on adhesive slides with VECTASHIELD HardSet Antifade Mounting Medium with DAPI (H-1500, Vector Lbs., USA). For IHC by DAB, sections were treated with 1:200 diluted biotinylated goat anti-rabbit antibody for 60 min (Vectastain ABC Elite Kit, Vector Lbs., USA), then with peroxidase conjugated avidin–biotin complex (Vectastain ABC Elite Kit, Vector Lbs., USA) according to the supplier’s protocol for 60 min. After rinses in PBS, the immunoreaction was developed in 1x TBS (pH 7.4) with 0.02% 3,3′ DAB (Merck KGaA, Germany) and 0.03 v/v% H2O2. The chemical reaction was carried out under visual control using a microscope to optimize the signal/background ratio, and was stopped after 10 min with 1x PBS. Then, preparations were washed with 1x PBS and mounted on gelatin-covered slides. After drying, slides were transferred into ascending ethanol solutions (50%, 70%, 96%, absolute 5 min, respectively), then into xylene for 2 × 10 min and cover slipped using Depex (Fluka, Germany) mounting medium. Brightfield and fluorescent images of CA1 and dentate gyrus (DG), according to Paxinos and Franklin [47] were acquired using a Nikon Eclipse Ti2 microscope (Nikon, Japan), then contrasted using Photoshop CS6 (Adobe, USA).